# Factorial structure, reliability, and construct validity of the Generalized Anxiety Disorder 7-item (GAD-7): Evidence from Malaysia

**Kai-Shuen Pheh**[1]\*, **Chee-Seng Tan**[1]\*, **Kai Wei Lee**[2], **Kok-Wai Tay**[1], **Hooi Tin Ong**[2], **Sook Fan Yap**[2]

**1** Department of Psychology and Counselling, Faculty of Arts and Social Science, Universiti Tunku Abdul Rahman (UTAR), Kampar, Malaysia, **2** Department of Pre-Clinical Sciences, Faculty of Medicine and Health Sciences, Universiti Tunku Abdul Rahman (UTAR), Sungai Long, Malaysia

\* phehks@utar.edu.my (KSP); cstan@outlook.my (CST)

**Data Availability Statement:** All relevant data used in analysis are available from the OSF repository (osf.io/jd5bt).

## Abstract

Generalized anxiety disorder (GAD) is one of the most common mental disorders in Malaysia. Psychometrically sound measurements are urgently needed to assess anxiety symptoms. The extensively used Generalized Anxiety Disorder 7-item (GAD-7) is a promising candidate. However, studies on its factorial validity show mixed findings. While the one-factor solution has been replicated in different cultural contexts, some studies found different factorial structures instead. This study aimed to clarify the factorial validity of the English version of the GAD-7 in the Malaysian context. The responses collected from 1272 emerging to older adults in Malaysia were randomly divided into two halves and submitted to exploratory factor analysis (EFA) and confirmatory factor analysis (CFA) respectively. Four acceptable models were explored in EFA ranging from unidimensional factor with 7 items to 3-factor models with 6 items. The four models revealed in EFA and the other competing models found in past studies were then examined and compared using CFA. The 6-item second-order model with a general factor of anxiety and three first-order factors with two items respectively (i.e., GAD-6) showed a more harmonic result and hence, is preferable. Moreover, the GAD-6 and its three subscales also showed satisfactory internal consistency and construct validity. This study uncovers a new and unique factorial structure of the GAD screening tool that fits in the Malaysian context. The scale may reveal GAD symptomatic dimensions that guide clinical interventions.

## Introduction

According to World Health Organization [1], anxiety disorder is one of the top 10 causes of disability-adjusted life year for 10 to 44 years old Malaysians in 2019. A scoping review of local studies on anxiety disorders published from 2005 to 2015 reported that the prevalence rate of anxiety disorders ranges from 1% to 67.6% among students, non-clinical and clinical populations due to the inconsistency in cut-off scores used in the studies [2]. Moreover, the

**Funding:** The data of this project was collected within the research project funded by Universiti Tunku Abdul Rahman Research Fund (Project No: IPSR/RMC/UTARRF/2020-C2/Y01) awarded to S. F. Yap.

**Competing interests:** The authors have no conflicts of interest to declare that are relevant to the content of this article.

disruptions associated with COVID-19 in the past two years have perpetuated poor mental health and resulted in 25.6% increase in anxiety disorder cases worldwide [3]. For instance, the prevalence risk of anxiety among 1851 university students in Malaysia was 29% [4]. In view of the current mental health landscape, there is an acute need for a psychometrically sound anxiety disorder measurement to facilitate treatment planning.

Generalized anxiety disorder (GAD) is a subtype of anxiety disorder. People with GAD tend to have chronic uncontrollable worry and excessive anxiety for no specific reason that occurs constantly for at least 6 months. The Generalized Anxiety Disorder-7 (GAD-7) was initially developed by Spitzer et al. [5] as a brief tool to identify probable diagnosis of generalized anxiety disorder and its severity. The initial version of the scale consisted of 13 items asking patients to rate how frequently they experience anxiety symptoms in the past two weeks. A simplified version with seven items with compatible psychometric qualities as the 13-item version was then developed. Since its introduction, GAD-7 has been used in different settings as a general measure of anxiety disorder and its severity [6, 7].

The GAD-7 has been validated across different countries and populations and has shown good psychometric properties. However, it is noteworthy that different factorial structures have been documented ranging from the original unidimensional structure, unidimensional structure with residual covariances, two-factor model with cognitive-emotional domain and somatic symptoms domain [6, 8], to higher-order model (see S1 Table for details). The inconsistency implies the GAD-7 is susceptible to cultural differences in the meaning of items [9].

## Overview of the present study

It is noteworthy that past studies have found different factorial structures of the GAD-7 in different cultural contexts. The inconsistency suggests that there could be cultural differences in the concept of generalized anxiety disorder. In other words, there is a possibility that GAD-7 may not capture the idea of generalized anxiety disorder in terms of the local perspective. Moreover, studies have found that some items of the measurements that have been validated in other contexts are not applicable to Malaysian samples [10–13], implying the items carry different meaning to Malaysian participants. Therefore, there is a possibility that the theoretical one-factor structure or some items of the GAD-7 may not hold in the Malaysian context. Employing the scale without verifying its factorial structure (and items) may result in inaccurate or incorrect findings.

In addition, although the Malay version of the GAD-7 has demonstrated good screening properties among women attending primary care clinics [14] and good factorial validity and reliability among diabetic outpatients in Malaysia [15], the psychometric qualities (e.g., factorial validity) of the English version has received less attention in the Malaysian context. It is inappropriate to assume that the psychometric qualities of the Malay version can be generalized to the English version. Moreover, English is a dominant medium of instruction in tertiary education institutions and industry sector in Malaysia [16]. The English version of the GAD-7 is widely used in clinical, research, and educational settings (e.g., [4, 17]). Hence, it is critical to understand if the scale possesses satisfactory psychometric properties in the Malaysian context. The present study aimed to address these issues by investigating the psychometric qualities of the English version of the GAD-7 in a Malaysian sample.

## Methods

### Participants

University students who were above 18 years old and studying at the randomly selected tertiary education centers in Klang Valley were recruited using convenience sampling. A total of 1272

university students ($M_{age}$ 21.50 SD = 3.10, range = 18 to 74) participated this study. The majority were female (72.2%), Malaysian (96.6%), and pursuing bachelor's degree (82.9%). There were 92.4% students living with family and almost one-third (32.7%) were from the bottom categories (B1/B2) of financial strata.

An online questionnaire was created using Google Forms and then distributed via Facebook, Instagram and WhatsApp to the university students in the Klang Valley. Data were collected between 4th to 17th January 2021 during the third wave of the COVID-19 Pandemic in Malaysia. Ethical approval was granted by the Scientific and Ethical Review Committee (Ref: U/SERC/211/2020). All respondents participated in the study voluntarily and gave their consent online (by checking the "I agree" box) prior to answering the online survey. The data of the present study are available at https://osf.io/jd5bt/?view_only= c92b739e563a44f3b8cdfaaf5ef7e448.

## Measurements

The GAD-7 is a self-report of the severity of generalized anxiety disorder and its symptoms with 7 items. Participants rated each item on a 4-point Likert scale ranging from 0 (Not at all) to 3 (Nearly every day). A total score was calculated by summing up the item scores. The score ranges from 0 to 21 and can be categorized as 0–4 (minimal anxiety), 5–9 (mild anxiety), 10–14 (moderate anxiety), and 15–21 (severe anxiety).

## Analytical approach

The data were randomly and equally divided into exploratory and confirmatory samples. The exploratory sample ($M_{age}$ = 21.42, $SD$ = 2.46; 472 women and 164 men) was submitted to exploratory factor analysis (EFA) with parallel analysis and principal axis factoring estimation using JASP (Version 0.16.2) [18] to explore the possible factorial structure of the GAD-7. Then, the confirmatory sample ($M_{age}$ = 21.65, $SD$ = 3.69; 447 women and 189 men) was submitted to confirmatory factor analysis (CFA) with diagonally weighted least squares (WLSMV) estimation. Statistical software R (4.2.0 for Windows) [19] and Lavaan R-package (Version 0.6–11) [20] were used to examine and compare the models revealed in the past studies and suggested by EFA and to identify the best factorial structure of the scale in the Malaysian context. A best-fit model shall fulfill the following criteria: comparative fit index (CFI) and Tucker-Lewis Index (TLI) $\geq$ .95, root mean square error of approximation (RMSEA) $\leq$ .05, standardized root mean square residual (SRMR) $<$ .08, and $\chi2/df < 3$. The expected cross-validation index (ECVI) was also assessed to assist in comparing alternative models with comparable goodness of fit. Specifically, a model with a smaller ECVI is preferable [21]. The composite reliability of the best fit model was examined using the compRelSEM function of the semTools R-package (Version 0.5–6) [22].

## Results

### Exploratory factor analysis

Parallel analysis with Promax rotation on the exploratory sample suggested a 3-factor solution. Supporting the factorability of the seven items, the Kaiser-Meyer-Olkin (KMO) test value was .911 and the Bartlett's test was statistically significant, $\chi^2$ (21) = 3131.330, $p < .001$. Although the model explained 72.90% of the total variance, the factor loading of item 2 exceeded 1.00, while item 3 had the lowest factor loading (.492). Therefore, we removed item 2 and reran EFA with the remaining six items. A 2-factor solution was revealed and explained 68.10% of the total variance. Items 1, 3, 6, and 7 loaded on the first factor while items 4 and 5 loaded the

second factor. The factor loadings ranged from .479 (item 1) to .917 (item 4). In addition, we conducted another EFA without item 3. A 3-factor solution was found and explained 73.80% of the total variance. The first factor consisted of items 4 and 5, the second factor consisted of items 1 and 2, and the last factor consisted of items 6 and 7. The factor loadings ranged from .488 (item 1) to .948 (item 2). Finally, we explored the original 7-item one-factor model by submitting and fixing all items to load on one factor. The one-factor solution was acceptable and explained 64.40% of the total variance. Table 1 shows the summary of the EFA results. In total, EFA revealed four potential factorial structures of the GAD-7. Further examination on these models using CFA is needed to clarify their fitness and superiority.

## Confirmatory factor analysis

We conducted several CFAs to identify the best fit model among the nine competing models that are revealed in the literature and our EFA. Specifically, we first examined the following five one-factor models: the original 7-item one-factor model (Model 1), 7-item one-factor model with a residual covariance between items 3 and 7 (Model 1a) [23], 7-item one-factor model with a residual covariance between items 1 and 2 and items 2 and 3 (Model 1b) [15], 7-item one-factor model with a residual covariance between items 4 and 5 and items 5 and 6 (Model 1c) [24], 7-item one-factor model with a residual covariance between items 4 and 5, items 5 and 6, items 4 and 6 (Model 1d) [7], and 7-item one-factor model with a residual covariance between items 1 and 2, items 4 and 5, items 5 and 6 (Model 1e) [25]. Next, we examined the following two-factor models: the 7-item two-factor model (Model 2) [26] and the 6-item two-factor model without item 2 suggested by EFA (Model 3). Then, instead of testing the 6-item three-factor model (without item 3) suggested by EFA, we tested the 6-item second-order model consisting of one general factor of anxiety and three specific first-order factors (Model 4) to account for the correlations among the three factors. Finally, we examined two 7-item second-order models. The first one consisted of one general factor of anxiety and three specific first-order factors (Model 5), which is a statistically equivalent form of the (7-item) 3-factor solution suggested by parallel analysis (in the present study). The second

**Table 1. Exploratory factor analysis results for the Generalized Anxiety Disorder 7-item.**

| Item | Model 1 | | | | Model 2 | | | Model 3 | | | | Model 4 | |
|---|---|---|---|---|---|---|---|---|---|---|---|---|---|
| | Factor 1 | Factor 2 | Factor 3 | Unique | Factor 1 | Factor 2 | Unique | Factor 1 | Factor 2 | Factor 3 | Unique | Factor 1 | Unique |
| 1 | 0.496 | | | .312 | 0.479 | | .364 | | 0.488 | | .304 | 0.828 | .314 |
| 2 | 1.015 | | | .077 | - | - | - | | 0.948 | | .096 | 0.837 | .299 |
| 3 | | 0.492 | | .313 | 0.659 | | .335 | - | - | - | - | 0.822 | .324 |
| 4 | | | 0.699 | .240 | | 0.917 | .160 | 0.843 | | | .208 | 0.834 | .304 |
| 5 | | | 0.872 | .238 | | 0.704 | .324 | 0.819 | | | .277 | 0.777 | .396 |
| 6 | | 0.783 | | .362 | 0.642 | | .415 | | | 0.527 | .430 | 0.740 | .453 |
| 7 | | 0.682 | | .355 | 0.871 | | .315 | | | 0.848 | .260 | 0.773 | .402 |
| SumSq. Loading | 1.748 | 1.726 | 1.628 | - | 2.209 | 1.877 | - | 1.772 | 1.388 | 1.265 | - | 4.507 | - |
| Proportion var. | .250 | .247 | .233 | - | .368 | .313 | - | .295 | .231 | .211 | - | .644 | - |
| KMO | .911 | | | | .897 | | | .887 | | | | .911 | |
| Bartlett's test | $\chi^2 (21) = 3131.330, p < .001$ | | | | $\chi^2 (15) = 2391.150, p < .001$ | | | $\chi^2 (15) = 2482.856, p < .001$ | | | | $\chi^2 (21) = 3131.330, p < .001$ | |

*Note.* N = 636. Factor loadings below 0.400 were omitted for the sake of clarity.

Unique = uniqueness; SumSq. Loading = sum of square loading, Proportion var. = proportion variance.

model suggested by Doi et al. [27] consisted of one general factor of anxiety and two specific first-order factors (Model 6).

Table 2 summaries the CFA results. Model 6 was not reported because standard errors were not computed as the model is not identified. The remaining models showed good fit except for Models 1, 1a, and 5. Among the remaining models, Model 4 is preferable because it fits relatively better than the other models. Specifically, Model 4 has the highest CFI and TLI values, the lowest RMSEA value, and the second lowest ECVI value. More importantly, the higher-order structure (with a general anxiety factor and three specific first-level factors) is conceptually aligned with the original scale and has better interpretability than the two-factor models. Therefore, the Generalized Anxiety Disorder scale with 6 items (hereinafter called GAD-6) is best represented by a second-order model in the Malaysian context. The GAD-6 consists of three first-order factors (see Fig 1): apprehensive expectation factor (items 1 & 2), autonomic excitation factor (items 4 & 5), and anxious distress factor (items 6 & 7). The unstandardized factor loadings were all statistically significant ($ps < .001$). The standardized factor loadings ranged from .786 (item 6) to .966 (the general factor to the autonomic excitation factor).

## Reliability and validity

Table 3 shows the descriptive statistics, inter-factor correlations (conducted using SPSS ver. 22), and compositive reliability for the GAD-6. The composite reliability values of the general factor and the three specific factors (i.e., subscales) were all greater than .760, supporting that the GAD-6 has acceptable to good reliability. As the GAD-6 consists of three specific factors,

**Table 2. The fit indices for the models of the GAD-7.**

| | Model | $\chi^2$ | df | p | $\chi^2$/df | CFI | TLI | RMSEA [90% CI] | SRMR | ECVI |
|---|---|---|---|---|---|---|---|---|---|---|
| 1 | 7-item one-factor model | 74.561 | 14 | < .001 | 5.33 | .968 | .952 | .083 [.065, .101] | .031 | .069 |
| 1a | 7-item one-factor model (with residual covariance between items 3 & 7) | 74.316 | 13 | < .001 | 5.72 | .968 | .948 | .086 [.068, .106] | .031 | .072 |
| 1b | 7-item one-factor model (with residual covariance between items 1 & 2 and 2 & 3) | 52.526 | 12 | < .001 | 4.38 | .979 | .963 | .073 [.053, .094] | .025 | .067 |
| 1c | 7-item one-factor model (with residual covariance between items 4 & 5 and 5 & 6) | 49.782 | 12 | < .001 | 4.15 | .980 | .965 | .070 [.051, .091] | .024 | .066 |
| 1d | 7-item one-factor model (with residual covariance between items 4 & 5, 5 & 6, and 4 & 6) | 40.957 | 11 | < .001 | 3.72 | .984 | .970 | .065 [.045, .087] | .021 | .066 |
| 1e | 7-item one-factor model (with residual covariance between items 1 & 2, 4 & 5, and 5 & 6) | 46.527 | 11 | < .001 | 4.23 | .981 | .964 | .071 [.051, .093] | .023 | .068 |
| 2 | 7-item two-factor model[a] | 43.040 | 13 | < .001 | 3.31 | .984 | .974 | .060 [.041, .081] | .022 | .061 |
| 3 | 6-item two-factor model[b] | 32.707 | 8 | < .001 | 4.09 | .984 | .970 | .070 [.046, .095] | .023 | .052 |
| 4 | 6-item second-order model[c] | 21.473 | 6 | .002 | 3.58 | .990 | .974 | .064 [.036, .094] | .017 | .053 |
| 5 | 7-item second-order model[d] | 57.185 | 11 | < .001 | 5.20 | .976 | .954 | .081 [.061, .103] | .026 | .072 |

*Note. N* = 636. The reported indices were based on robust values corrected in accordance with the WLSMV estimator. TLI = Tucker-Lewis Index, CFI = comparative fit index, RMSEA = root-mean-square error of approximation, CI = confidence interval, SRMR = standardized root mean square residual, ECVI = Expected Cross-Validation Index.

[a] Cognitive factor = items 1, 2, 3, & 7; somatic factor = items 4 to 6

[b] Factor 1 = items 1, 3, 6, & 7; Factor 2 = items 4 & 5

[c] Apprehensive expectation factor = items 1 & 2, autonomic excitation factor = items 4 & 5, and anxious distress factor = items 6 & 7

[d] Factor 1 = items 1 & 2; Factor 2 = items 3, 6, & 7; Factor 3 = items 4 & 5

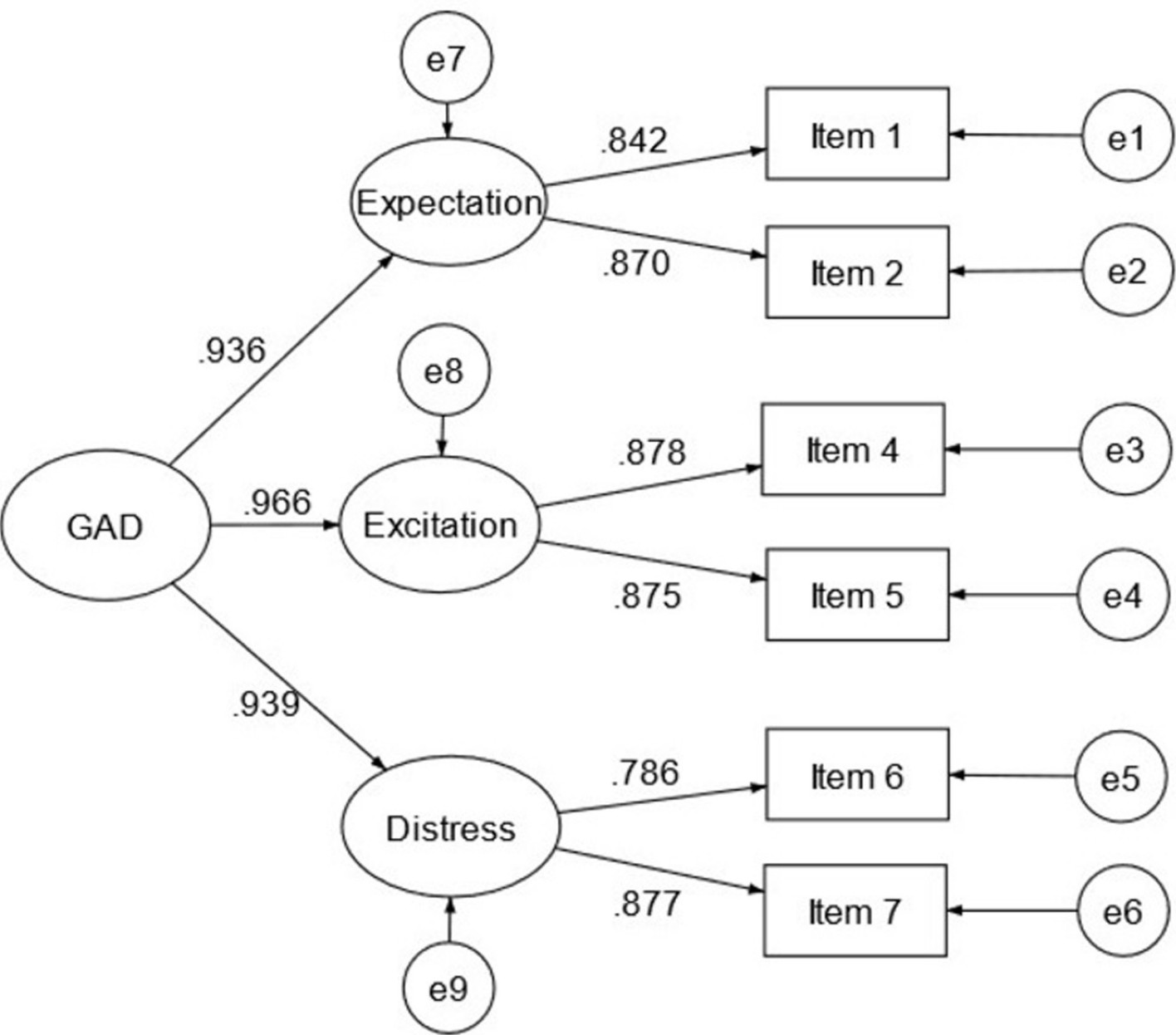

**Fig 1. Second-order model of the Generalized Anxiety Disorder 6-item (GAD-6).** Values shown are standardized parameter estimates. The item numbers shown here are the item numbers from the Generalized Anxiety Disorder 7-item. GAD = Generalized Anxiety Disorder; Expectation = Apprehensive expectation dimension; Excitation = Autonomic excitation dimension; Distress = Anxious distress dimension.

we conducted an unplanned examination of the convergent and discriminant validity of the GAD-6 using the average variance extracted (AVE) value. The three subscales showed adequate convergent validity because their AVE value was greater than .50 [28] respectively. Similarly, the three subscales also showed good discriminant validity. The AVE values of any two subscales were greater than the squared correlation coefficient between the two subscale scores. For example, the AVE values of the apprehensive expectation subscale (.733) and the

**Table 3. Descriptive statistics, composite reliability, intercorrelation coefficients, and average variance extracted for the GAD-6.**

| Factor | 1 | 1a | 1b | 1c | AVE |
|---|---|---|---|---|---|
| 1. GAD-6 | (.891) | .817 | .837 | .806 | NA |
| 1a. App. Expectation | .904*** | (.846) | .578 | .500 | .733 |
| 1b. Auto. Excitation | .915*** | .760*** | (.835) | .527 | .717 |
| 1c. Anx. Distress | .898*** | .707*** | .726*** | (.766) | .621 |
| Mean | 6.94 | 2.33 | 2.00 | 2.61 | NA |
| SD | 4.96 | 1.76 | 1.83 | 1.87 | NA |

*Note. N* = 636. AVE: average variance extracted; GAD-6: Generalized Anxiety Disorder 6-item (without item 3 of the Generalized Anxiety Disorder 7-item); NA: not applicable; App. Expectation = Apprehensive expectation subscale consisted of items 1 & 2; Auto. Excitation = Autonomic excitation subscale consisted of items 4 & 5; Anx. Distress = Anxious distress factor consisted of items 6 & 7; SD = standard deviation.

Composite reliability values were presented at the diagonal line. Pearson correlation coefficients were presented below the diagonal line, while the squared Pearson correlation coefficients were presented above the diagonal line.

*** $p < .001$

autonomic excitation subscale (.717) were greater than .578 (i.e., the squared correlation coefficient between the two subscale scores) respectively.

## Discussion

This is the first study that reveals a 6-item second-order model with three specific factors of the Generalized Anxiety Disorder (GAD-6). The new model is superior to the original and other competing models in a sample of Malaysian university students.

Our findings (of the second-order model) are consistent with Doi and colleagues' [27] discoveries of a higher-order model of the GAD-7. A key difference between the present study and Doi et al. [27] is that the latter merely performed CFA to verify the prior hypothesized models of the GAD-7. Considering the culturally sensitive nature of the GAD-7 [9] and the fact that the factorial validity of GAD-7 has not been established in Malaysia, we first employed EFA to explore the potential factorial structures followed by CFA to compare the competing models and identify the best fit model. This two-stage investigation allows us to discover other possible models that have not been identified in past studies and determine the superior model. As a result, this practice not only identified the factorial structure that fits the Malaysian context but also extends the literature by introducing the GAD-6, a new factorial structure of the GAD-7.

The GAD-6 and its three subscales have shown good reliability. The results are supportive that the GAD-6 may provide both clinically useful summed score and subscale scores. Following Stochl et al. [29], we recommend using the total score of the GAD-6 to indicate the severity level of individuals' anxiety disorder. Meanwhile, clinicians can refer to the subscale scores for gaining insights into clients' specific symptomatic dimensions [30] and then personalize the treatment plan to maximize therapeutic gains.

It was unexpected that Item 3 (*Worrying too much about different things*), which depicts an essential feature of generalized anxiety disorder [31], was not included in the best fit model. One of the possible reasons is the data were collected during the COVID-19 pandemic. Given the exceptional circumstances, worrying about various events or activities could be a normal reaction to the rapidly evolving pandemic instead of a sign of psychopathology.

Despite the promising findings, the present study is not without limitations. First, the results were derived from the original 7-item version of GAD-7. It is not clear whether the 6-item second-order model can be replicated if the data were collected using the newly

proposed GAD-6. Second, test-retest reliability and diagnostic validity of the GAD-6 are not examined. Finally, the GAD-6 has not been examined in other cultures and its generalizability is limited. Therefore, further studies with a more inclusive population spectrum are warranted to verify our findings and further investigate the psychometric qualities of the GAD-6.

## Conclusion

The Generalized Anxiety Disorder 6-item (GAD-6) with a second-order structure is a psychometrically sound tool for researchers and clinicians to assess the overall severity level of anxiety disorder, as well as symptomatic dimensions in the Malaysian context.

## Supporting information

**S1 Table. Summary the factor structure of the Generalized Anxiety Disorder 7-item.** (DOCX)

## Acknowledgments

The authors are grateful to the participants who took part in the study. We would also like to express our sincere gratitude to Kah-Yue Low, who played a crucial role in formatting this manuscript. Her tireless effort and dedication to ensuring the accuracy and consistency of the document were invaluable to the successful completion of this project.

## Author Contributions

**Conceptualization:** Kai-Shuen Pheh, Chee-Seng Tan, Kai Wei Lee, Hooi Tin Ong, Sook Fan Yap.

**Data curation:** Kai Wei Lee, Hooi Tin Ong, Sook Fan Yap.

**Formal analysis:** Chee-Seng Tan.

**Funding acquisition:** Sook Fan Yap.

**Investigation:** Sook Fan Yap.

**Methodology:** Kai-Shuen Pheh.

**Project administration:** Kai Wei Lee, Hooi Tin Ong, Sook Fan Yap.

**Writing – original draft:** Chee-Seng Tan, Kai Wei Lee, Kok-Wai Tay.

**Writing – review & editing:** Kai-Shuen Pheh, Kai Wei Lee, Hooi Tin Ong, Sook Fan Yap.

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
