## [Decision Letter · Decision Letter 0]

8 Mar 2023

PONE-D-23-03780Psychometric Properties of the GAD-7: Evidence from MalaysiaPLOS ONE

Dear Dr. Tan,

Thank you for submitting your manuscript to PLOS ONE. After careful consideration, we feel that it has merit but does not fully meet PLOS ONE’s publication criteria as it currently stands. Therefore, we invite you to submit a revised version of the manuscript that addresses the points raised during the review process.

An expert in the field of psychometrics has reviewed the paper and is believed the the contribution can add something to the literature. Therefore, I welcome the authors to revise their work according to the reviewer's comments. In addition, I would like the authors to further address the following two points.1. Please provide demographic information of the two separated samples (one for CFA and another for EFA) to let the readers know if the two samples are equivalent.2. Regarding the reviewer's comment on the use of English version in the present sample, please use the references of Gan et al., and Tung et al. to make justification.Gan, W. Y., Tung, S. E. H., Kamolthip, R., Ghavifekr, S., Chirawat, P., Nurmala, I., Chang, Y.-L., Latner, J. D., Huang, R.-Y., & Lin, C.-Y. (2022). Evaluation of two weight stigma scales in Malaysian university students: Weight Self-Stigma Questionnaire and Perceived Weight Stigma Scale. Eating and Weight Disorders, 27, 2595-2604.Tung, S. E. H., Gan, W. Y., Chen, J.-S., Kamolthip, R., Pramukti, I., Nadhiroh, S. R., Chang, Y.-L., Lin, C.-C., Pakpour, A. H., Lin, C.-Y., Griffiths, M. D. (2022). Internet-related instruments (Bergen Social Media Addiction Scale, Smartphone Application-Based Addiction Scale, Internet Gaming Disorder Scale-Short Form, and Nomophobia Questionnaire) and their associations with distress among Malaysian university student. Healthcare, 10, 1448.

We look forward to receiving your revised manuscript.

Kind regards,

Chung-Ying Lin

Academic Editor

PLOS ONE

“The authors have no conflicts of interest to declare that are relevant to the content of this article.”

6. Please include a separate caption for each figure in your manuscript.

7. Please upload a copy of Supporting Information Table S1 which you refer to in your text on page 5.

Reviewers' comments:

Reviewer's Responses to Questions

**Comments to the Author**

1. Is the manuscript technically sound, and do the data support the conclusions?

Reviewer #1: Yes

2. Has the statistical analysis been performed appropriately and rigorously? 

Reviewer #1: Yes

3. Have the authors made all data underlying the findings in their manuscript fully available?

Reviewer #1: No

4. Is the manuscript presented in an intelligible fashion and written in standard English?

Reviewer #1: Yes

5. Review Comments to the Author

Reviewer #1: I appreciate the opportunity to read this work and provide feedback. This manuscript attempted to examine the factor structure of GAD-7 in the Malaysian context with the use of EFA and CFA. The authors suggested a 6-item second-order model with a general factor of anxiety and three first-order factors (GAD-6). Overall, I believe the research has the potential to make a valuable contribution to the field. However, there are a few areas where I believe the manuscript could be strengthened.

1. I am curious about the reason for using the English version in this study. Specifically, the authors mentioned that "GAD-7 has demonstrated good screening properties among women attending primary care clinics in Malaysia (Mohd Sidik et al., 2012)", which was the Malay version. Therefore, please discuss your decision on using the English version of GAD-7 in investigating its factor structure in the Malaysian context.

2. Please provide more details about GAD-7 usage in Malaysia. Which language version, Malay or English, was typically employed for studies using the GAD-7 in Malaysia?

3. Is the study by Mohd Sidik et al. (2012) the only GAD-7 psychometric study done in Malaysia?

4. The authors mentioned that there is a Supplementary Table S1. However, I cannot find it. I am not sure if it is the fault of the system or if the author forgot to upload. I would appreciate it if the authors could upload it again.

5. ECVI was used to compare the models in this study, however, this is not mentioned in the analysis section. Please add the information.

6. Also, the descriptive statistics, inter-factor correlations, and the software to calculate them were not mentioned in the analysis section.

7. Regarding the subscales of the final GAD-6, the author named the three factors as expectation factor, autonomic excitation factor, and anxious distress factor. And later, the authors used the terms “cognitive subscale” and “somatic subscale”, which is confusing. I think discussion is needed for these names.

8. For the CFA, authors adopted factor structures from previous studies. Please add references for those models.

9. I think the manuscript title can emphasize factor structure.

6. PLOS authors have the option to publish the peer review history of their article (what does this mean?). If published, this will include your full peer review and any attached files.

Reviewer #1: No

---

## [Author Response · Author response to Decision Letter 0]

17 Apr 2023

Response to Academic Editor’s and reviewers’ comments

We are grateful to the reviewers and editors for their detailed and extremely thoughtful comments and suggestions for improvement. We have done our best to address every point made and highlighted the responses in green colour. 

Reviewer #1: I appreciate the opportunity to read this work and provide feedback. This manuscript attempted to examine the factor structure of GAD-7 in the Malaysian context with the use of EFA and CFA. The authors suggested a 6-item second-order model with a general factor of anxiety and three first-order factors (GAD-6). Overall, I believe the research has the potential to make a valuable contribution to the field. However, there are a few areas where I believe the manuscript could be strengthened.

1. I am curious about the reason for using the English version in this study. Specifically, the authors mentioned that "GAD-7 has demonstrated good screening properties among women attending primary care clinics in Malaysia (Mohd Sidik et al., 2012)", which was the Malay version. Therefore, please discuss your decision on using the English version of GAD-7 in investigating its factor structure in the Malaysian context.

Reply: English is a dominant language in Malaysia (The Malaysian Administrative Modernisation and Management Planning Unit, n.d.). We added two sentences to illustrate that the English version of the GAD-7 is also widely used in the clinical, research, and educational settings. However, the psychometric qualities of the English version have received less attention. Moreover, it is inadequate to assume that the good psychometric qualities of the Malay version of the GAD-7 can be generalized to the English version. See p. 5-6 for details.

The Malaysian Administrative Modernisation and Management Planning Unit. (n.d.). Official language. Retrieved March 12, 2023, from https://www.malaysia.gov.my/portal/content/30118

2. Please provide more details about GAD-7 usage in Malaysia. Which language version, Malay or English, was typically employed for studies using the GAD-7 in Malaysia?

Reply: Both Malay and English versions of the GAD-7 are widely used in Malaysia. The GAD-7 Malay version has been validated and used in healthcare and community settings in Malaysia (Sidik et. al, 2012; Maideen et al, 2015; Woon et al, 2020). The Malay version is mainly used in the public sectors (e.g., public hospitals, public schools) and contexts where Malay is the primary language medium (e.g., studies that targeting on Malay-speaking populations). 

On the other hand, the English version is commonly used in private sectors (e.g., private hospitals, private universities) and contexts where languages other Malay are used (e.g., studies that involve people whose mother tongue is not Malay). For example, GAD-7 English version was used in studies that involved university students (Irfan et. al, 2021; Mohamad, 2021). 

As both language versions are equally important, we do not think it is appropriate to indicate one is more typically employed than another.

References:

Sidik, S. M., Arroll, B., & Goodyear-Smith, F. (2012). Validation of the GAD-7 (Malay version) among women attending a primary care clinic in Malaysia. Journal of Primary Health Care, 4(1), 5-11.

Kader Maideen, S. F., Mohd Sidik, S., Rampal, L., & Mukhtar, F. (2015). Prevalence, associated factors and predictors of anxiety: a community survey in Selangor, Malaysia. BMC psychiatry, 15(1), 1-12.

Irfan, M., Shahudin, F., Hooper, V. J., Akram, W., & Abdul Ghani, R. B. (2021). The psychological impact of coronavirus on university students and its socio-economic determinants in Malaysia. INQUIRY: The Journal of Health Care Organization, Provision, and Financing, 58, 1-7. https://doi.org/10.1177/004695802110562

Woon, L. S., Hatta, S., & Norlaila, M. (2020). Factor Structure of The Malay-Version Generalized Anxiety Disorder-7 (GAD-7) questionnaire among patients with diabetes mellitus. Medicine & Health, 15(1).

Mohamad, N. E., Sidik, S. M., Akhtari-Zavare, M., & Gani, N. A. (2021). The prevalence risk of anxiety and its associated factors among university students in Malaysia: A national cross-sectional study. BMC Public Health, 21, Article 438. https://doi.org/10.1186/s12889-021-10440-5

3. Is the study by Mohd Sidik et al. (2012) the only GAD-7 psychometric study done in Malaysia?

Reply: To our best knowledge, Mohd Sidik et al. (2012) and Woon et al. (2020) are the two studies that focused on the evaluation of the psychometric qualities of the GAD-7 Malay version. Woon et al. (2020) study has been added on p. 5-6 and Table S1.

Several other studies reported internal consistency of the scale in their studies (Irfan et al., 2021; Maideen et al., 2015; ), but did not report other psychometric properties.

Malaysian Chinese Version of GAD-7 is available in the official website of GAD-7 (https://www.phqscreeners.com/) but its' psychometric properties data were not available to our best knowledge.

4. The authors mentioned that there is a Supplementary Table S1. However, I cannot find it. I am not sure if it is the fault of the system or if the author forgot to upload. I would appreciate it if the authors could upload it again.

Reply: Our apologies. The Supplementary Table S1, which summaries the factor structure of the Generalized Anxiety Disorder 7-item (GAD-7), has now been uploaded.

5. ECVI was used to compare the models in this study, however, this is not mentioned in the analysis section. Please add the information.

Reply: The following sentence was added to the manuscript (see p. 7) to indicate the use of ECVI in comparing alternative models.

The expected cross-validation index (ECVI) was also assessed to assist in comparing alternative models with comparable goodness of fit. Specifically, a model with a smaller ECVI is preferable (Browne & Cudeck, 1992).

6. Also, the descriptive statistics, inter-factor correlations, and the software to calculate them were not mentioned in the analysis section.

Reply: We indicated that the descriptive statistics and inter-factor correlations were computed using SPSS ver 22 on p. 10, rather than the analysis section, because the correlation analysis was conducted post-hoc after identifying a three-factor structure. 

7. Regarding the subscales of the final GAD-6, the author named the three factors as expectation factor, autonomic excitation factor, and anxious distress factor. And later, the authors used the terms “cognitive subscale” and “somatic subscale”, which is confusing. I think discussion is needed for these names.

Reply: We rectified the inconsistency. Below is the updated paragraph (on p. 10): 

The AVE values of any two subscales were greater than the squared correlation coefficient between the two subscale scores. For example, the AVE of cognitive the apprehensive expectation subscale (.733) and somatic the autonomic excitation subscale (.717) were greater than .578 (i.e., the squared correlation coefficient between the two subscale scores) respectively.

8. For the CFA, authors adopted factor structures from previous studies. Please add references for those models.

Reply: We inserted citations for the models as suggested. See p. 9 for details. 

9. I think the manuscript title can emphasize factor structure.

Reply: The manuscript title has been updated to Factorial Structure and Construct Validity of the Generalized Anxiety Disorder 7-item (GAD-7): Evidence from Malaysia.

 

Academic Editor

1. Please provide demographic information of the two separated samples (one for CFA and another for EFA) to let the readers know if the two samples are equivalent.

Reply: We added the demographic information of the exploratory sample (Mage = 21.42, SD = 2.46; 472 women and 164 men) and confirmatory sample (Mage = 21.65, SD = 3.69; 447 women and 189 men) on p. 7 in the Analytical Approach section.

2. Regarding the reviewer's comment on the use of English version in the present sample, please use the references of Gan et al., and Tung et al. to make justification.

Gan, W. Y., Tung, S. E. H., Kamolthip, R., Ghavifekr, S., Chirawat, P., Nurmala, I., Chang, Y.-L., Latner, J. D., Huang, R.-Y., & Lin, C.-Y. (2022). Evaluation of two weight stigma scales in Malaysian university students: Weight Self-Stigma Questionnaire and Perceived Weight Stigma Scale. Eating and Weight Disorders, 27, 2595-2604.

Tung, S. E. H., Gan, W. Y., Chen, J.-S., Kamolthip, R., Pramukti, I., Nadhiroh, S. R., Chang, Y.-L., Lin, C.-C., Pakpour, A. H., Lin, C.-Y., Griffiths, M. D. (2022). Internet-related instruments (Bergen Social Media Addiction Scale, Smartphone Application-Based Addiction Scale, Internet Gaming Disorder Scale-Short Form, and Nomophobia Questionnaire) and their associations with distress among Malaysian university student. Healthcare, 10, 1448.

Reply: Thanks for the suggestion. We found a Malaysian government website (see below) indicates that English is a dominant medium of instruction. Therefore, we cite the website when replying to the reviewer’s comment.

The Malaysian Administrative Modernisation and Management Planning Unit. (n.d.). Official language. Retrieved March 12, 2023, from https://www.malaysia.gov.my/portal/content/30118

---

## [Editor Report · Decision Letter 1]

24 Apr 2023

Factorial Structure, Reliability, and Construct Validity of the Generalized Anxiety Disorder 7-item (GAD-7): Evidence from Malaysia

PONE-D-23-03780R1

Dear Dr. Tan,

We’re pleased to inform you that your manuscript has been judged scientifically suitable for publication and will be formally accepted for publication once it meets all outstanding technical requirements.

Kind regards,

Chung-Ying Lin

Academic Editor

PLOS ONE

Additional Editor Comments (optional):

The authors have improved their contribution with the use of suggestions from the reviewer and I. Good job. 
---

## [Editor Report · Acceptance letter]

2 May 2023

PONE-D-23-03780R1 

Factorial structure, reliability, and construct validity of the Generalized Anxiety Disorder 7-item (GAD-7): Evidence from Malaysia 

Dear Dr. Tan:

I'm pleased to inform you that your manuscript has been deemed suitable for publication in PLOS ONE. Congratulations! Your manuscript is now with our production department. 

Kind regards, 

on behalf of

Dr. Chung-Ying Lin 

Academic Editor

PLOS ONE